# Stingless bee honey: Nutritional, physicochemical, phytochemical and antibacterial validation properties against wound bacterial isolates

**Miriam Wanjiru Mwangi** [1]*, **Tabitha W. Wanjau**[1], **Eric Omori Omwenga**[2]

**1** Department of Applied Sciences, School of Health Sciences, Kisii University, Kisii, Kenya, **2** Department of Medical Microbiology & Parasitology, School of Health Sciences, Kisii University, Kisii, Kenya

* mwangimirriam6@gmail.com

**Data Availability Statement:** All relevant data are within the manuscript.

**Funding:** The authors received no specific funding for this work.

## Abstract

With the rise of AMR the management of wound infections are becoming a big challenge. This has been attributed to the fact that most wound bacterial isolates have been found to possess various virulence factors like enzymes, toxins & biofilms production. Therefore, need for discovery of new lead compounds is paramount as such factors make these microbes to be resistant to already existing arsenal of antibiotics or even the immune system. This study aimed at documenting the nutritional, physicochemical, phytochemical and antibacterial properties of stingless bee honey. Isolation and characterization of bacterial isolates from 34 samples obtained from wounds of outpatients and surgical wards of Nakuru County Referral Hospital, Kenya was done. Various bacterial isolates (43) were isolated *Staphylococcus aureus* (34.8%) being predominant, followed by *Pseudomonas aeruginosa* (27.9%), *Klebsiella pneumoniae* (23.3%) and *Escherichia coli* (14.0%). A total of 36 out of the total isolates were genotypically characterized using molecular techniques detecting the prevalence of the following virulence genes; 16 srRNA (756 bp), hla (229 bp), cnf1 (426 bp), cnf2 (543 bp), hlyA (1011 bp), rmpA (461 bp), lasL (600 bp), gyrB (411 bp), khe (77 bp) and magA (128 bp). An assessment of the *in vitro* antibacterial activity of 26 stingless bee honey samples collected from their cerumen egg-shaped pots in Marigat sub-County, Baringo County, Kenya was done. Antibacterial properties of the stingless bee honey was done with varying susceptibility patterns being observed at different concentrations of honey impregnated discs ($10{\times}10^4$, $20{\times}10^4$, $50{\times}10^4$ and $75{\times}10^4$ ml μg/ ml) giving mean inhibition diameters of 18.23 ± 0.4 mm *(Staphylococcus aureus)*, 17.49 ± 0.3 mm *(Pseudomonas aeruginosa)*, 16.05 ± 0.6 mm *(Klebsiella pneumoniae)* and 10.19 ± 0.5 mm *(Escherichia coli)* with a mean range of 14.54 ± 2.0 mm to 17.58 ± 3 mm. Higher susceptibility to honey was recorded across all the bacterial isolates compared to conventional antibiotics while the mean MIC and MBC of the honey were recorded at 62.5 ml μg/ ml and 250 ml μg/ ml respectively. Control bacterial isolates *Staphylococcus aureus* ATCC 25923, *Escherichia coli* ATCC 25922, *Klebsiella pneumoniae* ATCC 27736 and *Pseudomonas aeruginosa* ATCC 27858 were used in the analysis. The stingless bee honey was found to be rich in various nutritive components like sugar (89.85 ± 5.07 g/100 g) and moisture (81.75 ± 10.35 mg/g) with a

**Competing interests:** The authors have declared that no competing interests exist.

significant difference of P <0.05 as the main antibacterial components. Additionally, the stingless honey did possess water soluble vitamins, proteins and minerals of which potassium was the most dominant one. In regard to phytochemicals, on our preliminary analysis phenolic, flavonoid and carotenoid compounds were found to be present with phenolic compounds being the most dominant one. Stingless bee honey from Marigat, has antimicrobial properties which could be attributed to the rich phytochemicals it possesses and its physicochemical properties in addition to its high nutritive value.

## 1. Introduction

Wounds represents disorder to an organ or a tissue involving devitalized and missing cellular structures and tissue layers [1] They may spread to other anatomical structures and the presence of proliferating microbes cause cellular hence inflammatory response enough to hinder the healing process [2]. In the United States chronic wounds affects between 1.5% and 3% of the general population, 15% to 20.3% in China, with about 14.8% overall incidences being reported in Sub-Saharan Africa, whereas in Kenya most injuries are implicated during drought seasons, intertribal crushes, cattle rustling as well as attacks by wild animals with the incidence rate of surgical site and post-caesarean wound infections of 7.0% and 19% respectively [3]. Wounds persistence in Kenya is enhanced by high treatment cost and high poverty levels as well as use of other medications like corticosteroids and chemotherapeutic [4].

Complicated wound and other infection management is caused by increase in emergence of drug and multidrug resistant microorganisms due to the overuse and misuse of antibiotics which normally cause toxic side effects to humans. [5] Furthermore, less new generation of antibiotics have been produced in the last decade with postulations that by 2050 we may be in a world with no active antibiotics [6] hence, need to explore nature for new remedial measures to curb the upsurge of the super resistant bugs. Despite all this, the local populations have been applying various traditional methods like medicinal plant use, honey use etc. to manage wounds for decades [7]. Since the ancient times humans have used honey in management of wound infections [8, 9]. Honey has also been found to possess antibacterial properties that do vary from one region to another based on previous studies [10]. These differences in terms of their antimicrobial activities have been attributed to differences in their chemical composition and physicochemical properties that have been found to differ according to the nectar origins, bee species and other intrinsic factors like osmotic effect, acidity, etc. [11]. Similar to the honeybees, the stingless bees collect nectar for honey production but store it in honey pots made of cerumen (contains the mandibular secretions) instead of hexagonal honeycombs attributed to the good quality of honey [12]. Based on this therefore, the current study aimed at deducing the antibacterial efficacy of honey from stingless bees that is commonly used by inhabitants of Baringo County, Kenya. The study further sort to validate the nutritional, physicochemical and phytochemical of the stingless bee honey.

## 2. Materials and methods

### 2.1 Study area

The study sampled the stingless bee honey from the semi-arid rangeland in Marigat Sub-County, Baringo County, Kenya that is a home for most acacia, low trees and shrubs (latitude 00˚26' to 00˚32'N, longitude 36˚00' to 36˚09E and altitude 900 to 1,200 M above the sea level,

agro-climatic zones IV &V, annual temp. & rainfall of 24.6˚C & 671 mm), according to Koppen and Geiger [13].

## 2.2 Sample size determination and sampling techniques

**2.2.1 Wound sampling techniques.** Wound swabs were obtained from outpatients as well as hospitalized surgical wards patients at County Referral Hospital- Nakuru with an informed patients' consent and established protocols as used before [14]. A swab was taken from every patient within inclusion criteria prior to being transported to the laboratory in Stuart transport medium. The sample size determination method for wound swabs employed for the target population was used as applied by [15] that ended up producing a sample size of 38 swabs as indicated below.

$$S = \frac{Z^2 X(P) X(1-P)}{C^2}$$

Where, Z–Z value (e.g. 1.96 for 95% confidence level)
P—Percentage picking a choice, expressed as a decimal
C—Confidence Interval

$$\text{Hence;} \frac{(1.96)^2 x(0.5)x(1-0.5)}{(0.025)^2} = 38$$

**2.2.2 Honey sampling techniques.** Stingless bees' hives were placed in a dark room and the honey from their honey pots removed by sucking with a syringe [16]. The sample size determination method for honey samples employed for the target population was as used by [17], and a sample size of 26 honey samples was obtained and used for this study as indicated below:

$$N = \frac{Z^2 XPXQ}{D^2}, \quad N = \frac{(1.96)^2 X\left(\frac{1500}{19895}\right) X 1 - P)}{(0.05)^2} = \frac{103}{4} = 26$$

## 2.3 Analysis of the physicochemical properties of the honey samples

Analysis was done as shown below.

**2.3.1 Determination of sugar content.** Spectrophotometric methods were used as previously applied [18]. Two grams of the honey sample was mixed with 10 ml dimethyl sulfoxide (DMSO) (25% v/v) and incubated in a water bath at 100˚C for 20 minutes. 0.5 ml of the mixture was diluted with 9.5 ml distilled water followed by addition of 0.5 ml of phenol (5%) and 2 ml of 75% sulphuric acid ($H_2SO_4$), finally absorbance was read at 492 nm against a standard glucose solution.

**2.3.2 Determination of moisture in honey samples.** The determination of moisture was done using a method described by [19]. Briefly, 2.0 grams of honey samples were weighed out and dried to a constant weight in a hot air oven at 70˚C and then cooled down and weighed again (oven-dry moisture content). The difference in the dry weight gave the moisture content.

**2.3.3 Determination of pH in honey samples.** To determine the pH of honey, a digital pH meter calibrated at pH 4 and 10 was applied as used before [20]. Briefly, working solution of honey was made by mixing 10g of honey in 75 ml of distilled water and using the calibrated pH electrode the pH reading on the meter was recorded.

**2.3.4 Determination of free acidity in honey samples.** The free acidity of the honey was determined using well established protocols as used before [21]. Ten grams of honey was dissolved in 75 ml of distilled water, the calibrated reference electrode was immersed and titrated to pH 8.5 with 0.05 M NaOH at a rate of 5 ml/minute and recorded. A blank determination procedure of 75 ml of distilled water was also run to pH 8.5, the free acidity is the acidity titrable with sodium hydroxide up to the equivalent point.

**2.3.5 Determination of hydrxymethylfurfural.** The spectrophotometric method involves measurement of UV absorbance of honey solutions as used before [20]. Five grams of honey were weighed and dissolved in 25 ml distilled water into a 50 ml volumetric flask. 0. l5 ml of Carrez solution I and 0.5 ml Carrez solution II were added and made up to 50 ml with distilled water, then filtered and after rejecting the first 10 ml of the filtrate the aliquots of 5 ml were put in two test tubes. 5 ml of distilled water and 5 ml of fresh 0.2% sodium metabisulphite were put into sample solution and reference solution tubes respectively. The absorbance was read at 284 nm and 336 nm using a UV-Visible mini ± 1240 Shimadzu spectrophotometer.

**2.3.6 Determination of hydrogen peroxide in honey.** This was done using a method used by [22]. Briefly, 30% (w/v) concentration of honey at pH8 by adjusting the pH metre was made followed by incubation in a water bath at 37˚C for 30 minutes. Hydrogen peroxide specific test strips were immersed into the solution for one second, the excess solution was shaken off the strip and the color developed was read against a color code to obtain the concentration of hydrogen peroxide in the honey samples.

## 2.4 Nutritive analysis of honey

**2.4.1 Determination of sugar content.** The sugar content determination was carried out using standard procedures as used by [16]. Briefly, 2.0g of the honey sample was mixed with 10 ml dimethyl sulfoxide (DMSO) (25% v/v) and incubated in a water bath at 100˚C for 20 minutes. 0.5 ml of the mixture was diluted with 9.5 ml distilled water, followed by 0.5 ml of phenol (5%) and 2 ml of 75% sulphuric acid ($H_2SO_4$) and the absorbance read at 492 nm against a standard glucose solution.

**2.4.2 Determination of honey protein concentration.** It was done according to Kjeldahl method as used by [23]. 0.5 grams of honey was transferred to the labelled kjeldahl tubes, with 2.5 grams of the catalytic mixture and 7 ml sulphuric acid. The mixture was placed in a block digester with gradual temperature increase (50˚C– 400˚C) for 5hours followed by heating the water to boil for 10 minutes and finally dissolved in 10 ml distilled water. To a 125 ml Erlenmeyer flask, 15 ml of boric acid (5%) was put followed by 5drops of the mixed indicator (Methyl red and Bromocresol green) which is red for acidic and green for basic, 20 ml of sodium hydroxide (50%) was then added to the digestion tubes until the samples were neutralized to a dark blue color. 50 ml of standard hydrochloric acid (0.01M) was put in a burette and titrated directly into the Erlenmeyer flask, pink color indicated endpoint and the volume of HCl used determined the protein concentration.

**2.4.3 Determination of Vitamin C content.** It was carried out using standard procedures as used before [24]. 100mg of honey sample was put into a flask followed by addition of 10 ml of metaphosphoric acid 1% and filtered through Whatman filter paper No. 4. Then 1 ml of the filtrate was mixed with 9 ml 2, 6- dichlorophenolindophenol (DCPIP) 0.005% and the absorbance analyzed spectrophotomically within 30 minutes at 515 nm.

**2.4.4 Determination of water—soluble vitamins in honey ($B_2$, $B_3$, $B_5$ and $B_9$).** The determination among the samples was carried out using the standard procedures as used before [25]. Briefly, 10 grams of the honey sample was dissolved through stirring in 10 ml distilled water followed by addition of 1 ml of sodium hydroxide (2 M), 12.5 ml of phosphate

buffer 1 M (pH5.5) and then topped up to the 50 ml mark in a volumetric flask with distilled water. The sample solutions were injected through the filter in a spectrophotometer at the selected wavelengths i.e. VitB2- 210 nm, VitB3–254 nm, VitB5–210 nm and VitB9–210 nm and the triplicate readings recorded.

**2.4.5 Determination of Calcium, Magnesium, Iron and Zinc in honey.** An Atomic Absorption Spectrometer (AAS) as used as before [26]. 10 grams of honey was heated to 500˚C to obtain a constant dry weight with an infrared lamp, in order to prevent foaming. The ash samples formed were dissolved in 10 ml perchloric acid (60%) and 10 ml nitric acid (65%), filtered and individual minerals determined using an air–acetylene flame and a hollow cathode lamp using known standards and wavelengths (Copper- 324 nm, Iron– 248 nm, magnesium- 285 nm and Zinc– 213 nm) for each individual mineral.

**2.4.6 Determination of sodium and potassium in honey samples.** The standard procedures as used before [27] were applied where 5 grams of honey was put into a 15 ml polypropylene flask then 0.4 ml of caesium chloride solution (5% m/v) and 10ul of hydrochloric acid were added followed by N–Propanol up to the 10 ml final volume. Internal standards were prepared by diluting ethanol and yttrium (1: 10 m/v) and absorbance measured out in the flame emission mode under different wavelengths (766 nm and 589 nm for potassium and sodium) specific for the minerals.

**2.4.7 Determination of Phosphorous in honey samples.** This was done in accordance with the described method as done before [27]. 10 grams of honey was put into a porcelain crucible and dissolved in 5 ml nitric acid (1 N) in tubes, heated in a water bath for 3 minutes followed by dissolving the mixture in distilled water up to 100 ml and filtered. 5 ml of the solution was put into 100 ml volumetric flask followed by addition of 10 ml of phosphate standard (0.1 mg/ml), 10 ml of nitric acid (6 N), 10 ml of ammonium molybdate (0.2%), and 10 ml of ammonium molybdate (5%) and finally diluted to the 100 ml mark with distilled water and allowed to stand for 15 minutes at room temperature to allow complete color development then absorbance was measured at 400 nm wavelength against a reagent blank for auto-zero and then recorded.

## 2.5 Determination of phytochemical and antioxidant properties of honey

**2.5.1 Total Content of Phenolic Compounds (TCPC).** This applied a method by [28], five grams of each honey sample was diluted to 50 ml with distilled water, mixed and filtered, followed by addition of 2.5 ml of 0.2N Folin- Ciocalteu and incubated at room temperature for 5 minutes. 2 ml of 75g/l sodium carbonate ($NaCo_3$) was then added and incubated again at room temperature for two hours in the dark and absorbance read at 760 nm against a methanol blank in spectrophotometer. Gallic acid was used as a reference standard and results expressed as mg gallic acid equivalent (mgGAE).

Flavonoids were determined by use of a method that was used before [29], 0.5 grams of honey was mixed with 5 ml of methanol (50%) and filtered, 5 ml of the honey solution was mixed with 5 ml of 2% aluminium chloride (AlCl3) and incubated for 30 minutes at room temperature and absorbance read at 420 nm using UV Visible spectrometer and expressed in milli grams of rutin equivalent (RE) in grams of honey.

The total content of carotenoids was determined by use of well-established protocol as used by [23]. 1 grams of the honey sample was mixed with 10 ml of n-hexane acetate mixture and homogenized thoroughly for 10 minutes at room temperature then allowed to stand in the dark for 30 minutes, filtered and absorbance of the filtrate measured at 450 nm using a spectrophotometer. The total carotenoids content was expressed as mg of β–carotene equivalents (mg β- carotene /kg of honey).

**2.5.2 Determination of Vitamin C (ascorbic acid) content.** The determination was by use of well-established protocol as used by [23]. To 100 grams of the honey sample, 10 ml 1% metaphosphoric acid was added at room temperature, filtered then1 ml of the filtrate was mixed with 9 ml 2,6, dichlorophenolindophenol (DCPIP) 0.005% and the absorbance measured within 30 minutes at 515 nm.

## 2.6 Quality assurance

Quality Control (QC) measures such as use of known standards, laboratory synthesized honey used as a negative control, the International Control Bacterial strains from American Type Culture Collection (ATCC), *Escherichia coli* (ATCC 25922), *Klebsiella pneumoniae* (ATCC 27736), *Pseudomonas aeruginosa* (ATCC 27853) and *Staphylococcus aureus* (ATCC 25923) were included in the study according to the Clinical and Laboratory Standards Institute (CLSI).

## 2.7 Isolation and characterization of wound microorganisms

**2.7.1 Phenotypic detection.** The suspected micro-organisms in the wound swabs were isolated through streaking on sterile agar plates for the identification and of pure microbial colonies. The evaluation of the morphological features together with Gram stain and biochemical tests (Coagulase, Methyl red, TSI and Oxidase tests) enhanced their further identification [30]. MacConkey and CLED agar were used in obtaining pure bacterial colonies which were then differentiated as Gram positive or negative upon staining.

**2.7.2 Molecular detection through characterization of specific virulence genes.** *i. DNA extraction and amplification*. This was done by use of the Fungi / Bacteria DNA premix kit from Zymo, USA, according to [31] protocol. The unbiased mechanical lysis of the microbes was achieved by bead beating with the ultrahigh density bashing beads then the debris were filtered from lysate using Zymo-spin. Washing followed to elute the solution from the Zymo-spin followed by complete PCR inhibitor removal by filtering the eluate with a Zymo-spin giving purified DNA ideal for microbial community profiling.

*ii. PCR- based screening of virulence genes*. This is a technique that makes multiple copies of a short sequence of target DNA by amplifying it in order to make enough material for detection and identification with a high degree of confidence by a fluorescence signal. DNA fragments of the pre-detected size were amplified by PCR primers with an amplification specific to the gene in question. Electrophoresis technique was used as before [32] to identify, quantify and purify nucleic acid fragments through loading the samples into wells on agarose and subjecting it to an electric field. The negatively charged nucleic acids move to the positive electrode, shorter fragments travel rapidly while longest fragments remain close to the origin of the gel resulting in separation based on size. Ethidium bromide DNA stain was used to enhance detection sensitivity while a 100 bp DNA ladder was used for quantitative analysis of linear double-stranded DNA in agarose gels by indicating the base pair (bp) length of nucleic acids. The virulent genes for detection specific to the bacterial isolates included *Staphylococcus aureus* (Hla and 16 srRNA), *Escherichia coli*, (CNF1, CNF2 and HlyA), *Klebsiella pneumoniae* (Khe, RmpA and MagA) and *Pseudomonas aeruginosa* (LasL and GyrB).

## 2.8 Antimicrobial activity determination of honey compared to conventional antibiotics

**2.8.1 Preparation of honey discs.** 50μl of various dilutions of stingless bees honey samples ($10 \times 10^4$, $20 \times 10^4$, $50 \times 10^4$ and $75 \times 10^4$ μg/disc) were impregnated on Whatman No. 1 filter paper discs (6 mm) and allowed to dry under sterile conditions.

**2.8.2 Disc diffusion method.** The disc diffusion (Kirby-Bauer) technique as used before [33] was employed using Muller- Hinton agar (Hi-Media, India), which was prepared according to the manufacturer's instructions. An inoculum prepared by emulsifying two colonies of the isolates in 10 ml distilled water and adjusting the turbidity to $1.5 \times 10^8$ CFU/ ml. (According to the McFarland Standards) was inoculated on Muller-Hinton agar through pour plating as done before [34, 35] after 15 minutes, the discs with test honey, reference and artificial samples were placed on the plates and incubated at 4˚C for 2hours to allow pre-diffusion of honey into the media, then at 37˚C for 24 hours and the diameter of each zone of inhibition measured. Cartridges containing commercially available antibiotics including Levofloxacin (5μg), Ampicillin (10μg), Tazobactum (110μg), Meropenem (10μg), Gentamycin (10μg) and Chloramphenicol (30μg) were also set against the bacterial isolates. Sterile Whatman No. 1 plain filter paper discs were also set on the plates as negative control.

**2.8.3 Evaluation of the bacteriostatic and bactericidal activity of honey.** Bacteriostatic and bactericidal properties of the honey samples were determined according with the micro dilution method as used by [36, 37]. 5 ml of nutrient agar was put in 8tubes, 5mg (5000μg) of honey was added to tube 1 and mixed well by vortexing to make a concentration of 1000 ml μg/ ml. This was serially twofold diluted using nutrient agar up to tube number 6 to obtain various ranges of concentration between 250 ml μg/ ml and 31.25 ml μg/ ml. Tube 7 was labelled GC and no honey was added. A volume of 100 ml μg/ ml of the bacterial inoculum was added into all tubes except tube 9 (HC) followed by incubation at 37˚C for 24hours followed by physical examination for turbidity. The tubes containing the least concentration showing no visible growth was considered as MIC and the previous lower dilution tube were inoculated onto sterile nutrient agar plates by the streak plate method and incubated aerobically at 37˚C for 24 hours. The least concentration that did not showed any growth was considered as the MBC.

## 2.9 Data analysis

The data in this study was first processed using IBM SPSS (Statistical Package for Social Sciences) statistics 23 and Two-way ANOVA through the Graph Pad Prism (Version 8.0.2–263) statistical software was used to compare mean values among the various experiments and to analyze their significant difference (P values < 0.005). Variables were considered significant by * and highly significant when P<0.01(**), P < 0.001 (***), P< 0.0001 (****).

## 2.10 Approvals

To ensure the required quality assurance, this study was approved by IREC (Ethics clearance for research proposal) and NACOSTI (National commission for science, technology and innovation).

## 3. Results

### 3.1 Characterization and identification of bacteria

**3.1.1 Culturing characteristics.** Various colonial morphologies were observed as; Large, round, convex, sharp border colonies with uniform golden–yellow color on CLED agar and those surrounded by clear zones of beta haemolysis on blood agar suggestive of *Staphylococcus aureus*, opaque, yellow lactose fermenting colonies with a slightly deeper yellow center on CLED agar suggestive of *Escherichia coli*, yellow to whitish–blue, extremely mucoid colonies on CLED agar as well as mucoid lactose fermenting colonies with a convex elevation on MacConkey suggestive of *Klebsiella pneumoniae* and colorless to pink, flat, irregular colonies with

alligator skin like surface and typical grape–like odour observed suggestive of *Pseudomonas aeruginosa*. A total of 28 burn and wound swabs (82.35%) yielded notable colonies of bacterial growth indicative of wound infection while 6 samples (17.6%) did not yield any colonies indicative of growth. According to the analysis, 15 (44.1%), 11(32.4%) and 2 (5.9%) wound swabs yielded 1, 2 and 3 isolates respectively hence giving rise to a total of 43 pure isolates as shown in S1 Table.

**3.1.2 Gram stain.** Various Gram reactions were realized from the isolated bacterial isolates as; Gram positive (purple) cocci shaped bacteria in clusters ("grape-like") suggestive of *Staphylococcus aureus*, Gram–negative pink-red rods (singly and in pairs) suggestive of *Escherichia coli*, Pink-red rod-shaped bacilli (singly, pairs, short chains and capsulated) suggestive of *Klebsiella pneumoniae* and Gram–negative rod-shaped bacilli that appeared slimmer and pale staining than members of *Enterobacteriaceae* as shown in supplementary data S2 Table.

**3.1.3 Biochemical reactions.** From the samples isolated in this study, 35% tested positive for coagulase test a possible positive indication of the presence of *Staphylococcus aureus*. When the samples were subjected to Triple sugar iron agar (TSI) 21% of the samples tested positive, an acidic slant (yellow) and an acidic butt (yellow) were observed indicating that the isolate was glucose, lactose and /or sucrose fermenter. Bubbles in the tube indicated production of carbon dioxide as a product of fermentation as a clear indication that the isolate could be *Escherichia coli as* shown on supplementary data S3 Table attached. Methyl red was also used to confirm presence of pathogens that do not ferment glucose. From the findings 46% of the isolates formed a red ring indicating a methyl red positive result, a finding that demonstrates the presence of *Escherichia coli*. Those that proved to be negative for this test (55%) was indicative of *Klebsiella pneumoniae* and finally, Oxidase test was done to differentiate *Pseudomonas aeruginosa* from other bacteria like *Escherichia coli and* it was deduced that 28% of the samples tested positive for the presence of *Pseudomonas aeruginosa* as indicated in supplementary data S3 Table.

**3.1.4 Molecular identification of bacteria through amplification of virulence genes.** In this study, molecular techniques (PCR assays and gel electrophoresis) were used to reveal the prevalence of 10 virulence genes including 16 srRNA (756 bp), hla (229 bp), cnf1 (426 bp), cnf2 (543 bp), hlyA (1011 bp), rmpA (461 bp), LasL (600 bp), gyrB (411 bp), khe (77 bp) and magA (128 bp) using appropriate primer sequences. According to supplementary data S1 Fig attached, a total of 36 out of 43 isolates (84%) indicated varying amplicons; 42% of *Staphylococcus aureus* (16 srRNA– 756 bp and hla– 209 bp—lane 1 and 2), 7% of *Escherichia coli* (cnf1- 498 bp, hly– 1177 bp), 17% of *Klebsiella pneumoniae* (magA—121 bp, rmpA 106 bp) and *Pseudomonas aeruginosa* (gyrB 222 bp and lasL 600 bp genes—lanes 12 and 13). Cnf2 and khe indicated no amplicons (lanes 4, 5, 8, 9, 10). Control organisms were also analyzed as a confirmatory for the presence of the virulence genes.

## 3.2 Physicochemical and nutritive properties of honey

The findings of physicochemical and nutritive properties were compared to the standards [38].

**3.2.1 Hydromethylfurfural (HMF).** A high HMF mean concentration (22.77 ± 13.6 mg/kg) was detected and was within the required range according to the International Regulations of quality [48] of 5 mg/kg to 80 mg/kg for honeys coming from tropical temperatures. There were significant differences between the samples (P<0.05) as shown in S2 Fig.

**3.2.2 Sugar concentration.** Mean sugar content of the stingless bee honey samples was 89.85± 6.0 g/100 g which was within the recommended values of 60 g/100 g to 800 g/100 g [39]. Samples from the arid semi-arid and medium altitude regions indicated varying mean

sugar concentrations of 94.0 ± 1.26 g/100 g and 86.0 ± 5.02 g/100 g. The values were significantly different across the regions (P<0.05) as indicated in S2 Fig.

**3.2.3 Moisture.** The stingless bee honey samples indicated a moisture mean content of 81.75 ± 10.4 mg/g which is in accordance as recommended by the [39] with a significant difference among of the honey samples across the different regions (P<0.05) as illustrated in S2 Fig.

**3.2.4 pH and free acidity.** The stingless honey samples had a mean pH value of 3.86 ± 0.11 and within the recommended limit (3.2 to 4.5) according to [39]. The mean free acidity was 0.063 ± 0.0, which was a bit low but within the recommended levels (<50 meq/kg) according to [38]. There were no significant differences in free acidity and pH between them (P>0.05) as indicated in S2 Fig.

**3.2.5 Hydrogen peroxide.** Hydrogen peroxide mean concentration of 1 ± 0.41 mM was recorded with highest mean concentration of 1.25 ± 0.35 mM recorded from Mukutani region and were within the recommended range of 0.5 and 2.5 mM according to [38]. There were no significant difference samples from different regions as indicated in S2 Fig.

**3.2.6 Water soluble vitamins.** Water soluble vitamins found in stingless bee honey include Vitamin B1, B2, B3, B5 and B9 (0.21 ± 0.03 to 0.64 ± 0.02 mg/l, 0.72 ± 0.03 to 2.39 ± 0.07 mg/l, 0.65 ± 0.5 to 4.19 ± 0.02 mg/l, 0.01 ± 0.0 to 1.22 ± 0.03 mg/l and 0.11 ± 0.00 to 1.64 ± 0.10 mg/l) which were within the Limit of Detection approved by the international Union of Pure and Applied Chemistry (1.10 mg/kg to 1.75 mg/kg). Vitamin $B_2$, $B_3$ and $B_9$ were significantly different (P<0.05) as in S3 Fig.

**3.2.7 Minerals.** The stingless bee honey indicated high mean levels of calcium, magnesium, iron, sodium and potassium as 2.11 ± 0.20 mg/l, 0.60 ± 0.47 mg/l, 0.76 ± 1.84 mg/l, 1.24 ± 0.95 mg/l and 17.94 ± 1.7 mg/l respectively which was lower than the recommended limit [44] of 3.23 to 236.8 mg/l, 4.85 to 218.0 mg/l, 2.18 to 563.72 mg/l, 16.66–1249.34 mg/l and 0.41–224.0 mg/l. There was significant difference (P<0.05) for Calcium, Magnesium and Zinc but non for Potassium, Iron, Sodium and Phosphorous as shown in S4 Fig.

**3.2.8 Crude proteins.** The honey samples contained detectable amounts of proteins giving an average value of 1.33 ± 0.89 g/100 g which were slightly higher than the recommended range of 0.9 to 0.5 g/100 g according to the [39]. The protein content of the honey samples were significantly different (p<0.05) as shown in S4 Fig.

## 3.3 Phytochemical properties of honey

The phytochemical components of stingless bee honey analyzed (Total flavonoid content, Total phenolic content, Vitamin C and Total carotenoid content) were expressed as mean range ± SD (mean average± SD) and the findings compared to the Standards of [38]. The honey samples indicated a higher mean Total Phenolic, Flavonoid and Carotenoid contents (92.18 ± 51.18 mgGAE/100 g, 23.7 ± 5.87mgRE/100g and 6.57 ± 0.21 mgβcarotene) as indicated in S5 Fig.

## 3.4 Antibacterial activity of honey samples and selected antibiotics against wounds and burns bacterial isolates

**3.4.1 Disc diffusion.** The antibacterial activity of stingless bee honey sample was compared with those of commonly used antibiotics (Levofloxacin (5 μg), Ampicillin (10 μg), Tazobactam (110 μg), Meropenem (10 μg), Gentamicin (10 μg) and Chloramphenicol (30 μg)), by use of sterile discs impregnated with varying concentrations ($10x10^4$, $20x10^4$, $50x10^4$ and $75x10^4$ μg/disc) of the stingless bee honey samples. The antibacterial potency of the honey samples was indicated by production of clear zones in concentrations of $50x10^4$ μg/disc and $75x10^4$ μg/disc with minimal inhibition or none for the concentrations of $10x10^4$ μg/disc and

20x10$^4$ μg/disc. *Staphylococcus aureus* was more susceptible followed by *Pseudomonas aeruginosa* then *Klebsiella pneumoniae* and finally, *Escherichia coli* shown in S4 Table and S6 Fig.

The study findings also did indicate that stingless bee honey had broad-spectrum antibacterial effect on the wound bacterial isolates. This was indicated by their susceptibility to the honey samples which compared well to the control bacterial isolates affirming its antibacterial effect against both Gram-positive and Gram-negative bacterial isolates. According to the findings of this study, the average inhibitions of all the stingless honey samples dilutions against the bacterial isolates compared well to the control bacterial isolates (ATCC) as tabulated in S5 Table.

The bacterial isolates recorded varying susceptibility to the different antibiotics under study at; 9.50 ± 0.23 mm to 25.0 ± 0.62 mm, 6.0 ± 0.0 mm to 24.0 ± 0.56 mm, 6.0 ± 0 .0 mm to 19.0 ± 0.50 mm and 6.0 ± 0.0 mm to 18.0 ± 0.46 mm for *Staphylococcus aureus*, *Escherichia coli*, *Klebsiella pneumoniae* and *Pseudomonas aeruginosa* respectively. *Staphylococcus aureus* showed the highest susceptibility against most of the antibiotics, but most isolates recorded considerable resistance except to Gentamicin (10μg) which indicated high inhibitory activity as indicated in S6 Fig.

### 3.5 Bacteriostatic and bactericidal activity of honey

The study evaluated the in vitro antibacterial activity (bacteriostatic and bactericidal effects) of stingless bee honey against four different bacterial cultures as wound isolates. The minimum inhibitory concentration for the honey samples was determined by micro dilution method indicating the inhibitory concentration at which stingless bee honey samples showed no visible growth of any test micro-organisms.

The MIC reflects the quantity needed for bacterial inhibition whose activity of the stingless honey samples against all the isolates was shown at 6.25 ml μg/ ml while the MBC activity was at 250 ml μg/ ml. The findings did show the honey samples have an optimum bacteriostatic activity on the test isolates showing its antibacterial activity at 125 ml μg/ ml as shown in S6 Table with *Escherichia coli* proving to be the most resistant isolate since it indicated the highest MIC activity at concentration of 125 ml μg/ ml and was killed completely at higher concentrations of 250 ml μg/ ml. Therefore, the honey's antibacterial activity (bacteriostatic and bactericidal effect) similar to antibiotics was ascertained.

## 4. Discussion

Microbial load in wounds cause delayed healing progress as they thrive well in hypoxic, non-healing wounds characterized by death and tissue necrosis [40]. This study identified 43 bacterial isolates from 34 wound swabs, *Staphylococcus aureus* was documented to be the most dominant isolate and was attributed to the fact that it forms the bulk of the normal skin flora followed by *Escherichia coli*, *Pseudomonas aeruginosa* and *Klebsiella pneumoniae* which was similar to other studies done by [41–45]. These bacterial isolates are among the most notorious wound infections causing bacterial strains upon entry into wound tissues. They can complicate illness, anxiety, increase patient discomfort, result in a lengthy hospital stay and can also lead to death [46]. Majority of them are facultative anaerobes and this makes them thrive well in wound environment that has less levels of oxygen supply hence can induce an infection [47].

The isolates obtained from this study were found to be in possession of various virulence genes. For instance, all 15 *Staphylococcus aureus* isolates expressed the Hla gene (that is commonly involved in inducing cytokines) in all clinical isolates a similar finding that was also documented by [48] on samples collected from children admitted in Upstate Golisano Children's hospital, New York. Production of cytokines assist in the virulence of *Staphylococcus*

*aureus* through modulation of the function of immune cells including neutrophils as well as triggering lysis of epithelial cells [49]. Further findings indicated that seven of the ten *Escherichia coli* isolates were carrying the cnf1 gene which helps the strain to escape from phagocytes as also documented by [50]. This assist this bacterium from being eliminated by the human body's phagocytic cells and the gene helps the bacteria to avoid this by provoking an increased adherence of polymorphonuclear leukocytes onto epithelial cells thus decreasing the rate of bacterial phagocytosis [51]. According to this study, the prevalence of magA and rmpA was reported to be 33% and 67% respectively and this was in agreement with the findings of [52] on molecular detection of capsular polysaccharide genes of Klebsiella pneumoniae on clinical samples collected from Hilla and Merjan Teaching hospital, Babylon. These genes therefore assist K. pneumoniae to produce capsules which have been found to protect them against phagocytic activities by the cells involved hence survival [53]. All *Pseudomonas aeruginosa* isolates in this study harbored lasL (600 bp) and gyrB (222 bp) a finding that is in tandem to the study findings on biopsy samples collected from Khatam-al-Anbia and Shahid Motahari hospitals, Iran by [54]. These genes assist *Pseudomonas aeruginosa* in quorum sensing that enhances their ability to form biofilms and produce other virulence traits like pyocyanin production which they commonly use to reduce ferric ions to ferrous state that is readily taken up as a nutritional requirement as well as reducing silver ions to the metallic state resulting in their lower concentration which is in consistent with its survival [55]. Contrary to these are the findings of [56] who detected 16 srRNA, ToxA and exoS virulence factor genes from wound and burn swab samples collected from Al-Diwaneyah hospital, Iran. These differences could be attributed to the various environs where the samples were picked from for the analysis. Presence of these virulence traits genes is a worrying finding as they clearly demonstrate how pathogenic and virulent these genes are and this can have a negative effect in their management strategies.

The physicochemical and nutritive components in this study are comparatively similar to previous reports for stingless bees in Oromia, Ethiopia [57] but contrary to our findings, a lower moisture content value (25–27 mg/g) was reported for Nigeria stingless bee honey [58], and higher moisture (199.0–419 mg/g) and lower pH (3.15–4.66) from stingless honey samples in Brazil [59]. This moisture content differences are due to climate, season and moisture content of the original plant nectar variations since at different temperatures honey dehydrates very fast. High moisture content is one of the causes of quality deterioration but when maintained at low levels it protects skin maceration by keeping the injured tissues moist as well as preventing growth of bacteria and other microorganisms [60]. According to our findings, higher mean moisture content were recorded by honey samples from Kibimgor and Koriema compared to Mukutani and Maoi due to the differences in altitude levels. Minerals in honey protects the tissues against free radical damage and from our study, all minerals were present in detectable amounts across all honey collection regions with potassium being the most abundant element. This is a consistent finding by other authors who considered this mineral to be the most quantitatively important in honey followed by calcium, sodium, iron and magnesium [61]. This indicates the botanical and geographical origins of the honey samples as well as contributing in antibacterial activity through enhanced electrical conductivity [62]. Water soluble vitamins are immune system stimulators and confer antimicrobial activity [63]. Our findings did indicate the presence of trace water soluble vitamins amount with very little mean value detected from Maoi and no detections made in Kibingor for Vitamin B5. This did differ with those of [64] who reported high values for vitB5 and VitB3 on Gelam and Acacia honey while other vitamins were not detected.

Phenols' antioxidant capacity result from the combined activity of flavonoids [65]. Our findings did indicate presence of low mean carotenoids, phenolics and flavonoids on samples

collected from Maoi while the values increased spontaneously for samples collected from Koriema region. These findings concurred with those reported by [66] from their study on Stingless bee in the Southern Malaysia, but contrary to these findings, higher Total Phenolic Content values (196 mgGAE/100 g)) and lower Total Flavonoid Content (18.511mgRE/100g) has been reported by [63, 67] respectively. Other findings of a study by [68] reported that Kelulut honey from East coast region showed highest values for phenolics (116.94 51.11 mgGAE/100 g), lower flavonoids (7.9 0.49mgQE/100g) and carotenoids (4.61 0.38m g/100 g). Our findings indicated that the analyzed honey samples contain adequate phenolic and flavonoid compounds to cause inhibition of the growth of pathogenic bacteria by disturbing the function of the cell membrane. These diversity in concentrations could be due to the differences in the geographical floral origins, specific climatic conditions and the types of bee species [69].

Based on our findings it was deduced that Gram-positive isolates exhibited larger zones of inhibition compared to Gram-negative isolates. These could be attributed to the fact that Gram positive isolates have a larger peptidoglycan layer moiety on the cell membrane and hence they are easily broken down by various antibacterial agents as compared to Gram negative isolates [70]. This finding is in tandem to a study by [68] on the antibacterial activity of stingless bee honey collected from Amazonas State against bacterial isolates. This could be attributed to the fact that stingless bee honey has been found to alter the outer membrane structure by changing the hydrophobic properties or mutations [55].

The MIC and MBC values in this study indicated that stingless bee honey samples have bactericidal and bacteriostatic activities potential against both control and wound isolates at reasonably low doses. Growth retardation and complete inhibition on all test organisms was observed at 62.5 to 250 ml μg/ ml with complete inhibition at very low concentration. This was similar to other studies conducted elsewhere [71–73]. Contrary to our findings is a study conducted by [74] in Ethiopia indicating that the growth of *Escherichia coli* and *Staphylococcus aureus* were completely inhibited at 650 ml μg/ ml while *Pseudomonas aeruginosa* was inhibited at 750 ml μg/ ml. This could be due to variations in cellular organizations as well as the variations in honey composition. It is from the findings of this study, that we recommend that stingless bee multifloral honey be considered for approval in therapeutic use.

## 5. Conclusion

Millions of chronic wounds continue to be diagnosed globally every year with evidence of bacterial infections facilitating long hospital stays, high treatment costs and further delay of wound healing. This could be due to the polymicrobial environment associated with these wounds which creates a favorable environment for the exchange of resistance genes between microorganisms. The antibacterial, antioxidant and medicinal properties such as anti-inflammatory and anti-cancer of stingless bee honey are highly dependent on its composition which depends on the climatic, environmental conditions as well as the diversity of the nectar composition of each producing plant species. The composition of stingless bee honey in this study indicates significantly great nutritional, physicochemical, phytochemical, antioxidant parameters and antimicrobial properties.

## Supporting information

**S1 Fig. Agarose rose gel (1%) electrophoresis.**
(PDF)

**S2 Fig. Physicochemical nutritive properties of stingless bee honey.**
(PDF)

**S3 Fig. Physicochemical nutritive properties of stingless bee honey.**
(PDF)

**S4 Fig. Physicochemical nutritive properties of stingless honey.**
(PDF)

**S5 Fig. Phytochemical nutritive properties of stingless honey.**
(PDF)

**S6 Fig. Antibacterial activity of commonly used antibiotics.**
(PDF)

**S1 Table. Cultural proportion of bacterial isolates.**
(PDF)

**S2 Table. Proportion of Gram staining reactions of bacterial isolates.**
(PDF)

**S3 Table. Biochemical tests.**
(PDF)

**S4 Table. Mean inhibition of the stingless bee honey.**
(PDF)

**S5 Table. Average inhibition of the stingless bee honey.**
(PDF)

**S6 Table. Minimum bacteriostatic and bactericidal concentrations.**
(PDF)

## Acknowledgments

1. William Kibyegon for his guidance through the vast Marigat Sub County.

2. Medical superintendent, County Referral Hospital, Nakuru for authorization of my access to the patients in the hospital.

3. Chief Medical Officer, Egerton University; Dr. G.K. Wahome for his authority the use of medical laboratory.

4. Laboratory technicians (Mr. Paul Kamau, Mr. Gerald Areba and Mr. Otieno), Biochemistry, Veterinary Sciences and Molecular Sciences laboratories, Egerton University.

5. Mr. Robert Kimutai for his enormous input in data analysis.

6. My entire family for your emotional and psychological support during this endeavor.

## Author Contributions

**Conceptualization:** Miriam Wanjiru Mwangi.

**Supervision:** Tabitha W. Wanjau, Eric Omori Omwenga.

**Writing – original draft:** Miriam Wanjiru Mwangi.

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
