## [Decision Letter · Decision Letter 0]

25 Jan 2023

PONE-D-22-28752Stingless bee honey: Nutritional, physicochemical, phytochemical and antibacterial validation properties against wound bacterial isolatesPLOS ONE

Dear Dr. mwangi,

Thank you for submitting your manuscript to PLOS ONE. After careful consideration, we feel that it has merit but does not fully meet PLOS ONE’s publication criteria as it currently stands. Therefore, we invite you to submit a revised version of the manuscript that addresses the points raised during the review process. As you will see, there are several small issues that must be addressed before the manuscript can go further in the publication proccess. Please, answer each point raised by the reviewers in your revised version of the manuscript. 

We look forward to receiving your revised manuscript.

Kind regards,

Tiago M. Francoy, Ph.D.

Academic Editor

PLOS ONE

and https://journals.plos.org/plosone/s/file?id=ba62/PLOSOne_formatting_sample_title_authors_affiliations.pdf.

5. We note that [Figure 1] in your submission contain [map/satellite] images which may be copyrighted. All PLOS content is published under the Creative Commons Attribution License (CC BY 4.0), which means that the manuscript, images, and Supporting Information files will be freely available online, and any third party is permitted to access, download, copy, distribute, and use these materials in any way, even commercially, with proper attribution. For these reasons, we cannot publish previously copyrighted maps or satellite images created using proprietary data, such as Google software (Google Maps, Street View, and Earth). For more information, see our copyright guidelines: http://journals.plos.org/plosone/s/licenses-and-copyright.

a. You may seek permission from the original copyright holder of Figure(s) [#] to publish the content specifically under the CC BY 4.0 license. 

Natural Earth (public domain): http://www.naturalearthdata.com/.

Additional Editor Comments:

The manuscript is interesting and deals with an important topic related to several aspects of stingless bees honey. However, reviewer 1 has raised several important points that must be addressed before the manuscript is ready for acceptance.

Reviewers' comments:

Reviewer's Responses to Questions

**Comments to the Author**

1. Is the manuscript technically sound, and do the data support the conclusions?

Reviewer #1: Partly

Reviewer #2: Yes

2. Has the statistical analysis been performed appropriately and rigorously? 

Reviewer #1: No

Reviewer #2: Yes

3. Have the authors made all data underlying the findings in their manuscript fully available?

Reviewer #1: No

Reviewer #2: Yes

4. Is the manuscript presented in an intelligible fashion and written in standard English?

Reviewer #1: Yes

Reviewer #2: Yes

5. Review Comments to the Author

Reviewer #1: PONE-D22-28752-REV-1

The authors clearly did a great job investigating diverse facets of honey. Their Ms needs more

clarity in the descriptions of the methods I started to read. Please, find attached initial comments in

the pdf. Basically: 1. Kindly review the plural of 'was' when needed, it should be ‘were’ in most of

the methods written in section 2.2. 2. The analytical units should be informed for all methods, not

only in some of them. 3. I am not familiar with the formula used to estimate 26 honeys. Is it necessary?

4. Figures 3-7 are blur, I would prefer Tables than graphs. 5. The stingless bee species identifications

are mandatory.

There is no entomological I.D. of the stingless bees. The authors report Maoi, Mukutani, Kibigor, and

Koriema Meliponin. It could be better meliponine or stingless bee. These locations are visible in

Figure 1 if you squeeze your eyes, authors can make them more visible, and inform them in the

abstract and the Materials and Methods section. Results in Figures 3-7 were classified for these four

locations, but the number of samples of each was not informed.

These names are in the Figures and in the Discussion, p. 18. They should be in the Materials and

Methods section, together with the stingless bee species identified by an entomologist, with a

vouchers deposited in a collection or museum. In the abstract, it is incorrect (Meliponines spp.) honey.

Baringo County had a recent review on ethnomedicinal uses of stingless bee honey (Kiprono et al.,

2022), so this Ms is worth to be published with more precise writing, and technical identification of

the stingless bees. It will be a good contribution for PLos One. Possibly they used one species of

each location? Or a unique stingless bee species? They could illustrate with an image of the

stingless bee nest, their entrances are a good feature to compare all of them.

The entomological origin of the honey is mandatory for this Ms. A great hurdle, but the authors need

that to complete this and their further research. Send about 10 dried bees to a global expert in Kansas

USA Michael S Engel msengel@ku.edu Another contact for that is in Belgium, the expert Alain

Pauly alain.pauly54@gmail.com. Seek for advice with them. Timothy Kegode tkegode@icipe.org is

writing a chapter on Kenyan stingless bees, he may help because he also faced difficulties with the

entomological I.D. of his stingless bee samples. There is not a SB expert entomologist in Kenya, and

custom clearance to send dried bees by courier is needed.

2.3 Analysis of the physicochemical properties of the honey samples Analysis was done in

accordance with [18] as shown below.

18. Bogdanov S, Jurendia T, Sieber R, Gallmann P. Honey for nutrition and health: A review.

Journal of the American College on Nutrition. 2009; 10 (4):745. 2008; 27(6): 677-689

This reference is a review on nutrition and health, not on analytical methods. Authors need to take

care of their references.

I am competent in chemical, biochemical, antioxidant activity of stingless bee honey, and the

importance to have the I.D. of their entomological origin. I also know melissopalynology and sensory

evaluation for these honeys, as well as biosurfactant activity. May I suggest the honey biosurfactant

test (HBT) could be added to this multiparametric Kenyan contribution of stingless bee honey? It will

need distilled water and diethyl ether. Interciencia, 2022 47 (10): 416-325.

https://www.interciencia.net/volumen-47-2022/volumen-47-numero-10/

Please, find attached my preliminary comments in the Ms pdf. Hope they help

Reviewer #2: I believe that the manuscript has the potential to be published because it addresses a very relevant subject and will add more to current knowledge, however, before being accepted, the authors must make some changes based on the manuscript analysis:

On page 17, second paragraph, where you say “high moisture content is one of the causes of quality deterioration…”. I suggest reviewing this information, because for stingless bee honey, the presence of high moisture and high acidity is one of its main characteristics and does not indicate deterioration of this honey. The indication of deterioration with high moisture is for honey from Apis mellifera bees.

I suggest observing Figures 1, 3, 4, 5 and 6 to improve the resolution and size of the graphics.

It is necessary to check the title of tables 1, 2 and 3 and standardize position, font size and place a more complete caption in all tables.

6. PLOS authors have the option to publish the peer review history of their article (what does this mean?). If published, this will include your full peer review and any attached files.

Reviewer #1: No

Reviewer #2: No

---

## [Author Response · Author response to Decision Letter 0]

2 Jul 2023

ALL THE POINTS RAISED DURING THE REVIEW PROCESS HAVE BEEN ADEQUATELY ADDRESSED TO CONFIRM TO THE REQUIRED PLOS ONE'S PUBLICATION CRITERIA

---

## [Decision Letter · Decision Letter 1]

11 Sep 2023

PONE-D-22-28752R1Stingless bee honey: Nutritional, physicochemical, phytochemical and antibacterial validation properties against wound bacterial isolatesPLOS ONE

Dear Dr. MWANGI,

Thank you for submitting your manuscript to PLOS ONE. After careful consideration, we feel that it has merit but does not fully meet PLOS ONE’s publication criteria as it currently stands. Therefore, we invite you to submit a revised version of the manuscript that addresses the points raised during the review process.

I have now received the comments of one of the original reviewers and also a new reviewer, since the other original one was not available. Both of the reviewers think that the results are important but pointed several points to improve the manuscript. I agree with the comments made by the reviewers, so, in face of the importance of the results, I suggest the authors to address the points raised by the reviewers before the manuscript can be accepted for publication in PLOS ONE.

We look forward to receiving your revised manuscript.

Kind regards,

Tiago M. Francoy, Ph.D.

Academic Editor

PLOS ONE

Reviewers' comments:

Reviewer's Responses to Questions

**Comments to the Author**

1. If the authors have adequately addressed your comments raised in a previous round of review and you feel that this manuscript is now acceptable for publication, you may indicate that here to bypass the “Comments to the Author” section, enter your conflict of interest statement in the “Confidential to Editor” section, and submit your "Accept" recommendation.

Reviewer #2: All comments have been addressed

Reviewer #3: (No Response)

2. Is the manuscript technically sound, and do the data support the conclusions?

Reviewer #2: Partly

Reviewer #3: Partly

3. Has the statistical analysis been performed appropriately and rigorously? 

Reviewer #2: Yes

Reviewer #3: No

4. Have the authors made all data underlying the findings in their manuscript fully available?

Reviewer #2: Yes

Reviewer #3: Yes

5. Is the manuscript presented in an intelligible fashion and written in standard English?

Reviewer #2: No

Reviewer #3: No

6. Review Comments to the Author

Reviewer #2: Abstract – AMR – do not abbreviate this right at the beginning – you should define abbreviations at the first appearance of the text.

Long abstract - Not exceed 300 words, as per the submission guidelines.

Paragraph starting at line 74 is extremely long. I recommend separating it into two points:

New paragraph in: “Since the ancient times humans have used honey in management …” page 80.

Another paragraph at: “Based on this therefore…” page 88.

Line 125 – Why is the title in italics?

Line 126 – Include the name and then the citation of the methodology used.

Line 135 – Protocol used before what? Suggestion: state that the protocol used is a protocol already established in accordance with... (insert a citation of the best method).

Line 142 – improve writing as instructed on line 135 about the method used.

Line 150 Include the name and then the citation of the methodology used.

Line 157 - Include the name and then the citation of the methodology used.

Line 166 – separate the sentences “10 minutes”

Line 167 – separate the sentences “5 drops”

Line 174 – orientation according to line 135

Line 178 – separate the sentences “30 minutes”

Line 180 – orientation according to line 135

Line 188 – separate the sentences “10 gram”

Line 202 – orientation according to Line 135 e separate the sentences “10 gram”

Line 204 – separate the sentences “3 minutes”

Line 208 – separate the sentences “15 minutes”

Line 219 – insert the methodology and then the citation

Line 221 – separate the sentences “30 minutes”

Line 233 – Missing period in sentence.

Line 225 – separate the sentences “1 gram”

Line 226 – separate the sentences “10 minutes”

Line 227 – separate the sentences “30 minutes”

Line 231 – orientation according to Line 135 and separate the sentences “100 grams”

Line 234 – separate the sentences “30 minutes”

Line 243 – microorganisms, without “–“

Line 250 – writing suggestion: This was done using the Fungi / Bacteria DNA premix kit from Zymo, USA, according to the protocol.

Line 280 – separate the sentences “15 minutes”

Line 281 – separate the sentences “2 hours”

Line 289 – separate the sentences “8 tubes”

Line 294 - separate the sentences “24 hours”

Between the lines 412 to 445 - Check the analytical units, because sometimes it appears together with the result and in others, separately. Example: “0.56mm, 6.0 ± 0 .0 mm”;

“50x104 µg/disc, 10x104µg/disc”

Part of the paragraph starting on line 245 is part of the discussion.

Line 457 - Revise all scientific names starting from the paragraph starting on Line 457 as they are not italicized. The same paragraph is too long. Break it down into at least 3 - 4 paragraphs.

Line 511 – separate “196mgGAE/100G” -> “196 mgGAE/100G”

Line 527 - Revise all scientific names starting from the paragraph starting on Line 527 as they are not italicized.

The resolution of the graphics in figures 2, 3 and 4 are not good.

It would be of great value to have the scientific name of the studied stingless bee species.

Reviewer #3: The research theme is really interesting and the findings are promising, but a full review of the English version as well as a full review of the graphs are mandatory. Please, find some suggestions and comments below:

Authors use abbreviations in all text, but the hole name to the expressions are missing (example: AMR – at the first line of the Abstract). Please write the full name before using na abbreviation (or provide a list of abbreviations).

METHODS

Item 2.4.1 “Determination of sugar content”: There is o need to repeat the method as described in 2.3.1. In this item, a comment that the same method will be used to describe the nutritive analysis of honey, concerning the sugar contente, is sufficient.

Some sentences have no meaning as some words are missing. Example: 2.7.1 Phenotypic detection: "... identification and of pure..." ???

Item 2.8.3: What are GC, HC, MIC, MBC and streak plate method?

2.9 Data analysis: The statement regarding the statistical softwares used is confusing. Authors used SPSS or Graph Pad Prism to analyse the data? Which post-hoc test was used after ANOVA? This is extremely important to analyze the graphs presented in the figures.

“(P values <0.005)”??? I imagine that authors mean P < 0.05. Also if atuhors put a limit for the minimum difference (P < 0.05), the others are also significant and not "highly significant".

RESULTS

In order to be in agreement with the Methods' oganization, authors should first describe and present the physicochemical, phytochemical and nutritional characteristics of the honey before describing the bacteria identification and the antibiotic properties of the honey.

3.1.2 – Attention to pontuation.

3.1.3 Biochemical reactions: There are results obtained with methods that were not described in the “Methods” section: coagulase test, triple sugar iron agar and oxidase test.

3.2.2: There are 3 regions citted (arid, semi-arid and médium altitude) and only 2 sugar concentrations described.

3.2.3 and others: The regions showed in graphs were not citted in Methods. The first time the regions' names appear is in the Figure 3 legends. The regions from where the honey samples were collected must be named in the item “2.1 Study area”, in the Methods' section.

3.2.4 pH and free acidity: “(P>0.05). Where there is no statistical significance, please write the precise value of P.

3.2.5 Hydrogen peroxide: First sentence: This sentence made no sense. Is there something missing? Please re-write it.

3.2.7 Minerals: Instead of “non for Potassium”, write “not for Potassium”.

3.3 Phytochemical properties of honey: Last sentence: What are these unities: mgGAE/100g and mgRE/100g? They should be described in the methods' section.

3.4.1 Disc diffusion, 3rd paragraph: “The bacterial isolates recorded varying susceptibility to the different antibiotics under study at; 9.50...” This sentence is totally nuclear. Please rewrite.

In the same paragraph: “...but most isolates recorded considerable resistance except to

Gentamicin (10µg)...”. This parto f the sentence is not clear. Please rewrite.

DISCUSSION

There are many erros that need to be corrected. Please revise the English version of this part. Some examples are below:

Last paragraph of page 16: “assist in the virulence” Do authors mean: “contribute to the virulence”?

FIGURES

Figures are unclear. Which kind of post-hoc test was done? Tukey-Kramer? Dunnet?

The "stars" on the histograms indicate diference between (or among) which data?

Examples:

(a): stars indicate difference among what data?

(b): all histograms have stars! So, they are compared with what?

(e): The first histogram is the control group? So, the stars must be placed over the other bars.

(a) and (f): why the histograms have no bars?

The legend in each graph has the same name as the graph's title. Please take the legend off, as it is not necessary (and it is difficult to read).

Similar comments could be applied to Figures 4 to 7. Please revise the statistical analysis and reconstruct the graphs.

7. PLOS authors have the option to publish the peer review history of their article (what does this mean?). If published, this will include your full peer review and any attached files.

Reviewer #2: No

Reviewer #3: No

---

## [Author Response · Author response to Decision Letter 1]

1 Dec 2023

THANK YOU FOR YOUR COMMENTS. I HAVE GONE THROUGH EACH REVIEWER'S COMMENTS AND HAVE ADDRESSED THEM. WE ARE READY TO CONFIRM TO THE STANDARD OF PLOS ONE

---

## [Decision Letter · Decision Letter 2]

12 Feb 2024

PONE-D-22-28752R2Stingless bee honey: Nutritional, physicochemical, phytochemical and antibacterial validation properties against wound bacterial isolatesPLOS ONE

Dear Dr. MWANGI,

Thank you for submitting your manuscript to PLOS ONE. After careful consideration, we feel that it has merit but does not fully meet PLOS ONE’s publication criteria as it currently stands. Therefore, we invite you to submit a revised version of the manuscript that addresses the points raised during the review process.

We look forward to receiving your revised manuscript.

Kind regards,

Marcello Iriti, Ph.D.

Academic Editor

PLOS ONE

Additional Editor Comments:

Please provide a detailed letter (point by point) to the reviewer 3 comments

Reviewers' comments:

Reviewer's Responses to Questions

**Comments to the Author**

1. If the authors have adequately addressed your comments raised in a previous round of review and you feel that this manuscript is now acceptable for publication, you may indicate that here to bypass the “Comments to the Author” section, enter your conflict of interest statement in the “Confidential to Editor” section, and submit your "Accept" recommendation.

Reviewer #2: (No Response)

Reviewer #3: (No Response)

2. Is the manuscript technically sound, and do the data support the conclusions?

Reviewer #2: Yes

Reviewer #3: (No Response)

3. Has the statistical analysis been performed appropriately and rigorously? 

Reviewer #2: Yes

Reviewer #3: (No Response)

4. Have the authors made all data underlying the findings in their manuscript fully available?

Reviewer #2: Yes

Reviewer #3: (No Response)

5. Is the manuscript presented in an intelligible fashion and written in standard English?

Reviewer #2: Yes

Reviewer #3: (No Response)

6. Review Comments to the Author

Reviewer #2: The authors failed to make the corrections described in the previous review, regarding the standardization of measurement units (mass, temperature), mainly in materials & methods, lacking space between the value and the measurement unit. Please refer to the previously submitted detailed review and make corrections.

Reviewer #3: The authors only sent a "Thank you message" but did not correct or justify the majority of points that were addressed. In this way, the manuscript continue to be in low quality to be published in PlosOne.

7. PLOS authors have the option to publish the peer review history of their article (what does this mean?). If published, this will include your full peer review and any attached files.

Reviewer #2: No

Reviewer #3: No

---

## [Author Response · Author response to Decision Letter 2]

4 Mar 2024

THE CORRECTIONS ON THE UNITS AND SPACING IN MATERIALS AND METHODOLOGY SECTION HAVE BEEN MADE AS STATED

---

## [Editor Report · Decision Letter 3]

13 Mar 2024

Stingless bee honey: Nutritional, physicochemical, phytochemical and antibacterial validation properties against wound bacterial isolates

PONE-D-22-28752R3

Dear Dr. MWANGI,

We’re pleased to inform you that your manuscript has been judged scientifically suitable for publication and will be formally accepted for publication once it meets all outstanding technical requirements.

Kind regards,

Marcello Iriti, Ph.D.

Academic Editor

PLOS ONE

Additional Editor Comments (optional):

The authors have satisfactorily answered questions from reviewers 2&3, MS can be accepted.
---

## [Editor Report · Acceptance letter]

29 Apr 2024

PONE-D-22-28752R3 

PLOS ONE

Dear Dr. MWANGI, 

I'm pleased to inform you that your manuscript has been deemed suitable for publication in PLOS ONE. Congratulations! Your manuscript is now being handed over to our production team.

Kind regards, 

on behalf of

Prof. Marcello Iriti 

Academic Editor

PLOS ONE